# Study on Phyllosphere Microbial Community of Nettle Leaf during Different Seasons

Shuan Jia [1,2,†], Yongcheng Chen [1,†], Rongzheng Huang [1] , Yuxin Chai [1], Chunhui Ma [1,*] and Fanfan Zhang [1,*]

1   Grass Land Science, College of Animal Science and Technology, Shihezi University, Shihezi 832000, China; 2015230408@stu.shzu.edu.cn (S.J.); 20202013022@stu.shzu.edu.cn (Y.C.)
2   Institute of Animal Health Supervision of Xinjiang, Urumqi 830011, China
*   Correspondence: chunhuima@126.com (C.M.); zhangfanfan@shzu.edu.cn (F.Z.)
†   These authors contributed equally to this work.

**Abstract:** Nettle (*Urtica cannabina*) is an excellent feed resource widely distributed worldwide. Phyllosphere microbes are important as they have living conditions similar to those of the above-ground parts of host plants. Exploring amino acids (AA) and microorganisms can further understand the growth of plants in different seasons. The present study investigated the content of AA and phyllosphere microbes' structure of nettle plants in different seasons. The results found that AA contents varied significantly with the season, such as alanine, aspartate, cysteine, glutamate, glycine, and methionine contents decreased significantly from spring to winter ($p < 0.05$), the contents of arginine, histidine, serine, and lysine were highest in summer ($p < 0.05$). The results suggested that the diversity of bacteria and fungi both increased during winter. During winter, Sphingomonas (relative abundance 25.22–28.45%) and Filobasidium (27.6–41.14%) became dominant. According to the redundancy analysis (RDA) of the correlation between AA and microbes, these two microbes were both the most important factors and showed a negative correlation with AA during winter. Thus, seasons could significantly affect the distribution of phyllosphere microbial communities on the nettle, especially in winter. According to the function prediction(PICRUS2 (KEGG pathway) and FUNGuild) results, the bacteria in the phyllosphere of *U. cannabina* mainly participated in metabolism. Pathogenic fungi were relatively high in autumn. The present study reveals the influence of seasonal change on the phyllosphere microbial community in *U. cannabina*.

**Keywords:** amino acid; nettle; phyllosphere microbial; bacteria community; fungal community

## 1. Introduction

Nettle is an annual or perennial herb in the genus *Urtica* in the family *Urticaceae*, which is distributed widely in many parts of the globe, including North Africa, parts of Asia, Europe, and North America [1]. According to the 'Flora of Xinjiang,' Nettle (*Urtica cannabina*) is distributed widely in Xinjiang, China. It mainly grows in the warm desert grassland, between the low-altitude areas in the north and south of the Tianshan Mountains and at the edge of the Gobi. Nettle could provide a source of a high-quality diet for grazing livestock in natural grassland due to its richness in vitamins and trace elements and its high nutritional value. In addition, it has fast growth, high drought tolerance, and high medicinal value [2,3].

Currently, due to the gradual degradation of grassland ecological environments, natural grassland environments face various challenges, including low soil fertility, salinization, drought, and large temperature fluctuations, which greatly affect the normal growth of plants [4]. In addition, seasonal changes have a great relationship with plant diseases [5]. Such challenges limit the exploitation of plant resources to a great extent. Amino acids (AA) are precursors of proteins and other organic compounds closely associated with plant nutrition and play active roles in plant responses to various stress factors [6,7]. AA accumulation

in plants facilitates abiotic stress resistance [8]. In addition, AAs are intermediate products of some metabolic pathways and act as signaling molecules that regulate various metabolic, physiological, and biochemical pathways and, in turn, numerous physiological processes in plants [9,10]. Further, AA deficiency is a major cause of plant nutrient imbalance [11]. Therefore, AA content is a key indicator of plant health and growth and facilitates tolerance to various abiotic and biotic stresses, especially microbes in the environment.

Numerous previous studies have shown that rhizosphere microbes influence plant physiology and enhance biotic or abiotic stress resistance by facilitating numerous activities and processes, such as nitrogen fixation, nutrient uptake, and synthesis of bioactive compounds in host plants [12–14]. There had co-occurrence networks between the rhizosphere and microbial phyllosphere of plants—the habitat composed of the above-ground effective parts of plants is collectively called interlobar, and the microorganisms living on its surface and inside are called interlobar microorganisms [15]. Consequently, phyllospheric microbes play a key role, mainly because they have living conditions similar to those of the above-ground parts of host plants and networks with other microbes from different plant parts [16]. Furthermore, AAs participate in biogeochemical cycles and could influence plant biology and ecological function [17], such as the bioremediation of harmful substances [18] and the protection of plant hosts from pathogens [19]. Thus far, studies focused on nettle growth, such as physiological and photosynthetic characteristics, none of them had studies on the microbial community of nettles [20,21]. In addition, under changing living environments, such as season, temperature, humidity, and ultraviolet light intensity [22,23], phyllosphere microbes could establish unique plant–microbe symbiosis systems through various strategies and interactions with plants [24,25]. A review showed that AA metabolism is closely associated with plant–microbe interactions, providing signaling molecules, nutrients, and defense compounds [26]. However, the relationships between AA and phyllosphere microbes during nettle growth remain unclear. Specifically, it is unclear which AA participates in disease resistance and which AA attracts beneficial phyllosphere microbes. Therefore, the present study aimed to preliminarily investigate the correlations between AA contents and phyllosphere microbial community structure in nettle under seasonal variation. The results of the present study could enhance our understanding of plant–microbe symbiosis systems.

## 2. Materials and Methods

### 2.1. Overview of the Study Area

The study site was located in the temperate desert–steppe type rangeland of *Seriphidium* sp., in Shawan County, Tianshan Mountains, Xinjiang (E 84°58′–86°24′ N 43°26′–45°20′, Elevation 1100–1300 m). The local area has a temperate desert arid climate, abundant sunshine, and heat, an annual average precipitation of 185 mm, and an annual temperature of 6.9 °C. The average temperature of the hottest month (July) is 25.77 °C, and the maximum temperature is >40 °C. The frost-free period is 170–190 day. The soil is typical chestnut soil.

The observation area was established within an area fenced over the long term. The vegetation mainly comprised *Seriphidium transiliense*, *Stipa* spp., *Festuca valesiaca* subsp. *sulcata*, and *Carex liparocarpos*. The associated species were *Kochia prostrata*, *Agropyron* spp., *Leymus* spp., *Ceratocarpus arenarius*, *Trigonella arcuata*, *Gagea* sp., and *Tulipa* sp., with other short-lived plants. *U. cannabina* was mainly distributed in the grassland environment (Figure 1).

### 2.2. Materials Preparation

Nettle samples were collected from the study area. The sampling times were 10 May (spring), 10 July (summer), 10 September (autumn), and 10 November (winter) in 2019. Each sampling is from 11: 30–12:00 a.m. Temperature and humidity during sampling times were obtained by use temperature and humidity recording instrument (testo 625, Hainan, China) for intervals of 5 min during 11: 30–12:00 a.m., which were 20.3 °C and 3.1% for 10 May (spring), 28.5 °C and 26.93% for 10 July (summer), 16.32 °C and 10.19%

for 10 September (autumn), and 1.27 °C and 62.14% for 10 November. The twenty plant samples were randomly selected, and leaf surfaces were scraped gently with sterile surgical blades (To remove the influence of other factors on the sample). The scraped leaves were placed randomly in 50-mL sterilized centrifuge tubes (three replicates per period, 5–8 g per replicate, with 12 samples in total). At the same time, all the scraped leaves were collected and stored in liquid nitrogen. All samples were transported to the laboratory immediately for further preservation (−80 °C) and analyses.

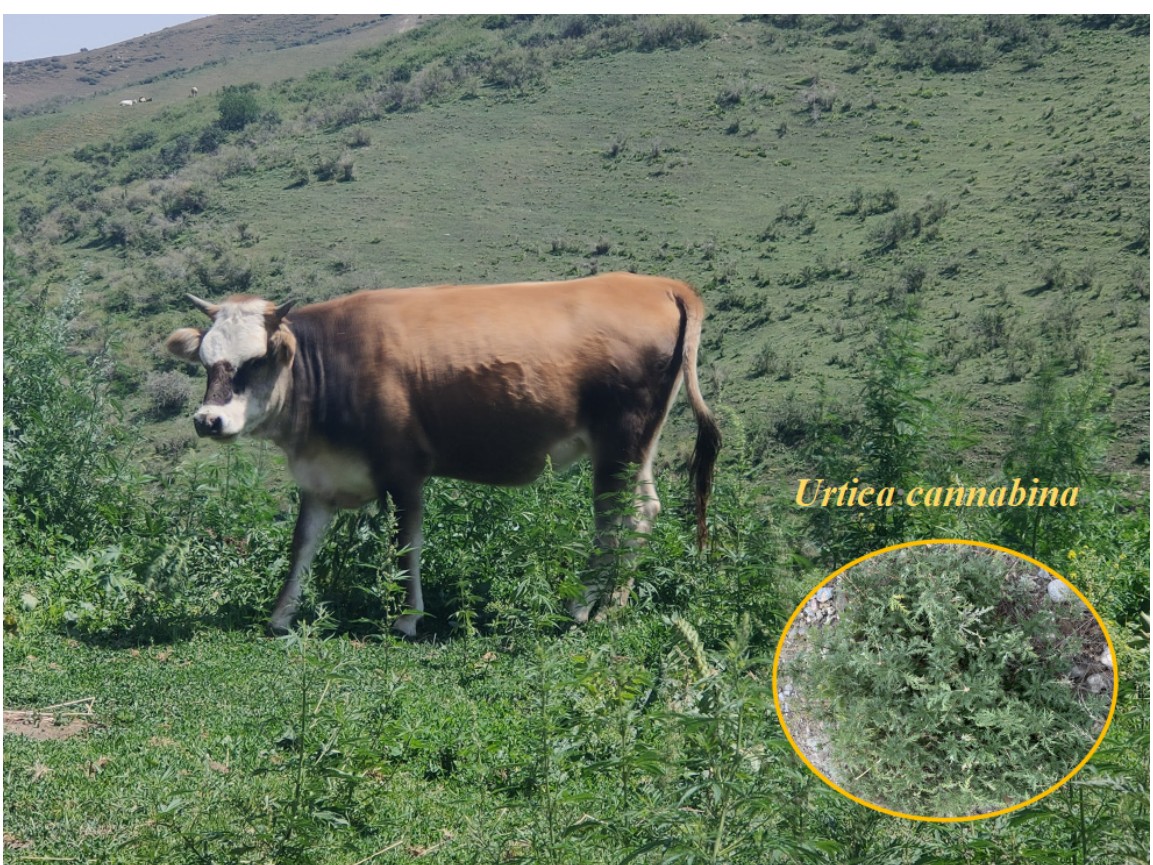

**Figure 1.** Nettle (*Urtica cannabina*) growth environment.

*2.3. Amino Acids Content Analysis*

The AA contents (%) were determined using an AA analyzer (LA8080, Hitachi Ltd., Tokyo, Japan). Briefly, the sample was homogenized and hydrolyzed. Tryptophan was hydrolyzed with 4.2 mol NaOH, and the remaining AA was hydrolyzed with 6 mol HCl [27]. The chromatographic column was a Hitachi #2622PH 4.6 mm I.D. × 60 mm. The ammonia removal column was a Hitachi #2650L 4.6 mm I.D. × 40 mm. The mobile phase was Hitachi MCL buffer accessories (a mixture of 18 types of AA standard solution). The derivation reagent was Hitachi ninhydrin chromogenic solution. The measurement parameters were 440 nm and 570 nm (detection wavelength), 0.40 mL/min (buffer flow rate), 0.35 mL/min (derivation reagent flow rate), 57 °C column temperature, and 135 °C derivation temperature. The injection volume was 20 mL, and each treatment was measured in triplicate.

*2.4. Bacterial and Fungal Community Sequencing Analysis*

Total microbial genomic DNA was extracted from nettle samples using the E.Z.N.A.® soil DNA Kit (Omega Bio-tek, Norcross, GA, USA). DNA quality and concentration were determined using 1.0% agarose gel electrophoresis and a NanoDrop® ND-2000 spectrophotometer (Thermo Scientific Inc., Waltham, MA, USA) and kept at −80 °C prior to further

use. The V3–V4 regions of the bacterial 16S rRNA gene were amplified using the 338F (5′-ACTCCTACGGGAGGCAGCAG-3′) and 806R (5′-GGACTACHVGGGTWTCTAAT-3′) primer pair in an ABI GeneAmp® 9700 PCR thermocycler (ABI, CA, USA). The primers targeting the ITS1 regions (ITS1F: 5′-CTTGGTCATTTAGAGGAAGTAA-3′; ITS2R: 5′-GCTGCGTTCTTCATCGATGC-3′) of fungi were used to conduct PCR amplification. The PCR reaction mixture, including 4 μL 5 × Fast Pfu buffer, 2 μL 2.5 mM dNTPs, 0.8 μL each primer (5 μM), 0.4 μL Fast Pfu polymerase, 10 ng of template DNA, and ddH2O to a final volume of 20 μL. PCR amplification cycling conditions were as follows: initial denaturation at 95 °C for 3 min, followed by 27 cycles of denaturing at 95 °C for 30 s, annealing at 55 °C for 30 s and extension at 72 °C for 45 s, and single extension at 72 °C for 10 min, and end at 4 °C. All samples were amplified in triplicate. Purified amplicons were pooled in equimolar amounts, and pair-end sequenced on an Illumina MiSeq PE300 platform/NovaSeq PE250 platform (Illumina, San Diego, CA, USA) according to the standards recommended by Majorbio Bio-Pharm Technology Co., Ltd. (Shanghai, China). OTUs are based on 97% sequence identity. The raw sequencing reads were deposited into the NCBI Sequence Read Archive (SRA) database (Accession Number: bacteria, PRJNA828128; fungi, PRJNA828139).

## 3. Statistical Analysis

The AA data were subjected to a one-way analysis of variance. Data were analyzed in IBM SPSS Statistics 21 (IBM Corp., Armonk, NY, USA). Significant differences between treatments were determined using Tukey's test at $p < 0.05$. Bioinformatic analysis of nettle plants was carried out on the Majorbio cloud platform (https://cloud.majorbio.com accessed on 12 January 2022). Briefly, based on the Operational Taxonomic Unit (OTU) information, rarefaction curves, and alpha diversity indices, including observed OTUs, Chao1 richness, Shannon index, and Good's coverage (Alpha diversity analysis can reflect the abundance and diversity of microbial communities and the greater Chao1 index, the higher the richness of microbial communities. The greater the Shannon index value, the higher the microbial community diversity in the sample), were calculated using Mothur v1.30.1 [28]. Principal Coordinate Analysis (PCoA) was conducted based on Bray–Curtis dissimilarity using the Vegan v2.5-3 package. Permutational multivariate analysis of variance was used to assess the percentage of variation explained by the treatments and the statistical significance using the Vegan v2.5-3 package. Further analysis was carried out using the Majorbio Cloud platform.

## 4. Results

### 4.1. Amino Acid Contents of Nettle Leaves under Seasonal Variation

As shown in Table 1, the AA contents in nettles were affected significantly by season. Alanine, aspartate, cysteine, glutamate, glycine, and methionine contents decreased significantly from spring to winter ($p < 0.05$). Arginine, histidine, serine, and lysine contents were in the order of spring > summer and autumn > winter ($p < 0.05$). Isoleucine, leucine, and valine contents were in the order of summer > spring and autumn > winter ($p < 0.05$). Proline and phenylalanine contents were in the order of summer > autumn > spring > winter ($p < 0.05$). Tyrosine contents were in the order of summer > spring > autumn and winter ($p < 0.05$). Threonine contents were in the order of summer > spring > autumn > winter ($p < 0.05$). Tryptophan contents were the highest in summer ($p < 0.05$). Generally, AA contents in spring and summer were higher than in autumn and winter.

### 4.2. Phyllosphere Microbe Composition across Seasons

Phyllosphere microbial diversity (bacterial and fungi) in nettle was determined using high-throughput sequencing analysis. A total of 1,640,454 (bacterial) and 2,000,344 (fungi) optimized sequences were obtained, and the effective sequences were clustered into 2524 (bacterial) and 2693 (fungal). Figure 2A–C (bacterial) and Figure 2I–III (fungi) show the Shannon and Chao index dynamics. Winter and autumn had the highest bacterial diversity, and winter had the highest fungal diversity ($p < 0.05$). Bacterial and fungal diversity was

higher in spring than summer ($p < 0.05$). Bacterial richness was the highest in winter, and fungal richness was the lowest in winter ($p < 0.05$). In addition, autumn had the highest fungal richness ($p < 0.05$); however, there were no significant differences in bacterial richness among spring, summer, and autumn.

**Table 1.** Amino acid contents in nettle leaves across seasons.

| Amino Acids (AA) % | | Spring | Summer | Autumn | Winter | SEM | *p* Value |
|---|---|---|---|---|---|---|---|
| Non-essential AA | Alanine | 1.83 [a] | 1.26 [b] | 0.92 [c] | 0.51 [d] | 0.15 | *** |
| | Arginine | 2.18 [a] | 1.56 [b] | 1.57 [b] | 1.04 [c] | 0.12 | ** |
| | Aspartate | 2.11 [a] | 1.82 [b] | 1.55 [c] | 1.24 [d] | 0.10 | *** |
| | Cysteine | 1.29 [a] | 1.10 [b] | 0.94 [c] | 0.62 [d] | 0.07 | *** |
| | Glutamate | 1.86 [a] | 1.61 [b] | 1.32 [c] | 0.91 [d] | 0.11 | *** |
| | Glycine | 0.82 [a] | 0.71 [b] | 0.64 [c] | 0.49 [d] | 0.04 | *** |
| | Histidine | 1.20 [a] | 0.88 [b] | 0.64 [b] | 0.51 [c] | 0.08 | ** |
| | Proline | 1.68 [c] | 2.12 [a] | 1.79 [b] | 1.00 [d] | 0.12 | *** |
| | Serine | 2.13 [a] | 1.75 [b] | 1.77 [b] | 1.04 [c] | 0.12 | ** |
| | Tyrosine | 1.13 [b] | 1.22 [a] | 0.03 [c] | 0.02 [c] | 0.17 | ** |
| Essential AA | Isoleucine | 0.51 [b] | 0.62 [a] | 0.42 [b] | 0.21 [c] | 0.05 | ** |
| | Leucine | 2.12 [b] | 2.62 [a] | 2.00 [b] | 1.34 [c] | 0.14 | ** |
| | Lysine | 1.94 [a] | 1.59 [b] | 1.62 [b] | 1.15 [c] | 0.09 | ** |
| | Methionine | 1.38 [a] | 1.21 [b] | 1.04 [c] | 0.82 [d] | 0.06 | *** |
| | Phenylalanine | 1.70 [c] | 2.48 [a] | 2.04 [b] | 1.01 [d] | 0.16 | *** |
| | Threonine | 0.97 [b] | 1.18 [a] | 0.83 [c] | 0.42 [d] | 0.08 | *** |
| | Tryptophan | 0.42 [b] | 0.79 [a] | 0.35 [b c] | 0.29 [c] | 0.06 | *** |
| | Valine | 0.11 [b] | 0.30 [a] | 0.12 [b] | 0.06 [c] | 0.03 | *** |

Means of three observations. SEM: standard error of mean. Different lowercase letters in the same line indicate significant differences ($p < 0.05$). ** $p < 0.01$, *** $p < 0.001$.

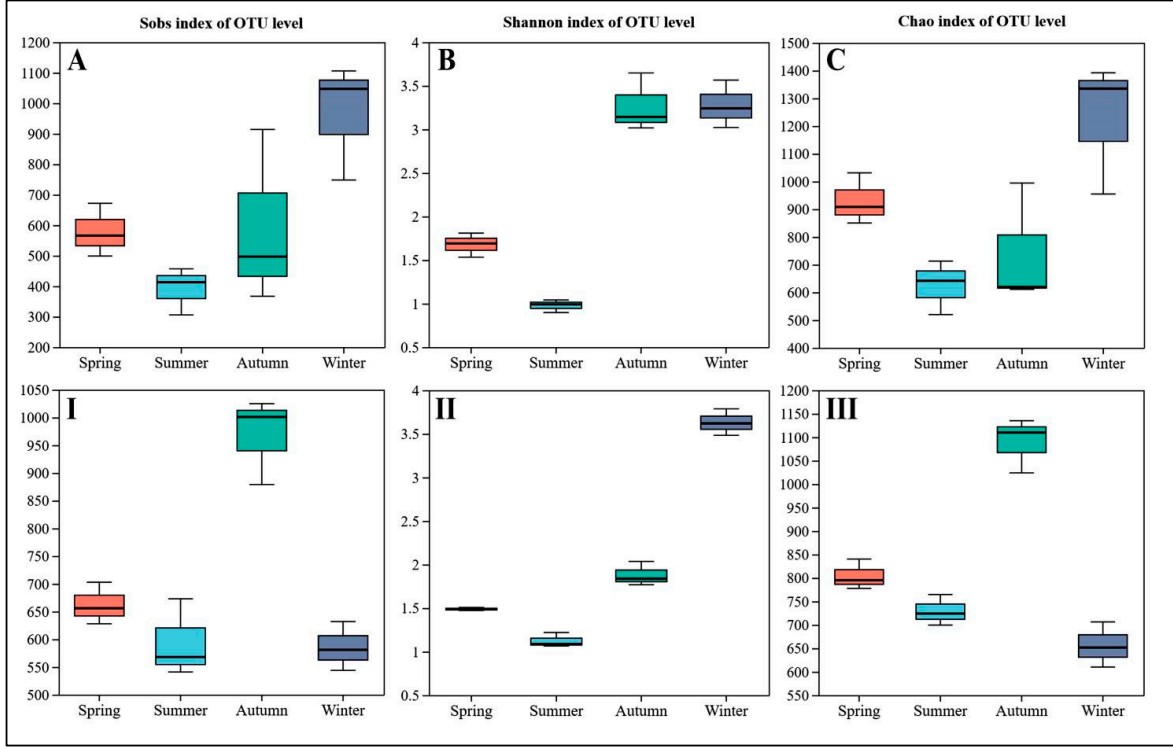

**Figure 2.** Alpha diversity of phyllosphere microbes in nettle. Capital letters indicate bacterial community. Roman numerals indicate fungal community. (**A,I**), Observed species; (**B,II**), Shannon index; (**C,III**), Chao index. Season was the sampling time. Spring, 10 May; Summer, 10 July; Autumn, 10 September; Winter, 10 November.

In the present study, Principal Coordinate Analysis (PCoA) results based on Bray–Curtis distance indicated that season affected the distribution of phyllosphere microbial taxa significantly (ANOSIM, *p* < 0.001, Figure 3A,B) in nettle. Season could explain 89.07% of the variance in bacterial community structure (Figure 3A) and 94.58% of the variance in fungal community structure (Figure 3B).

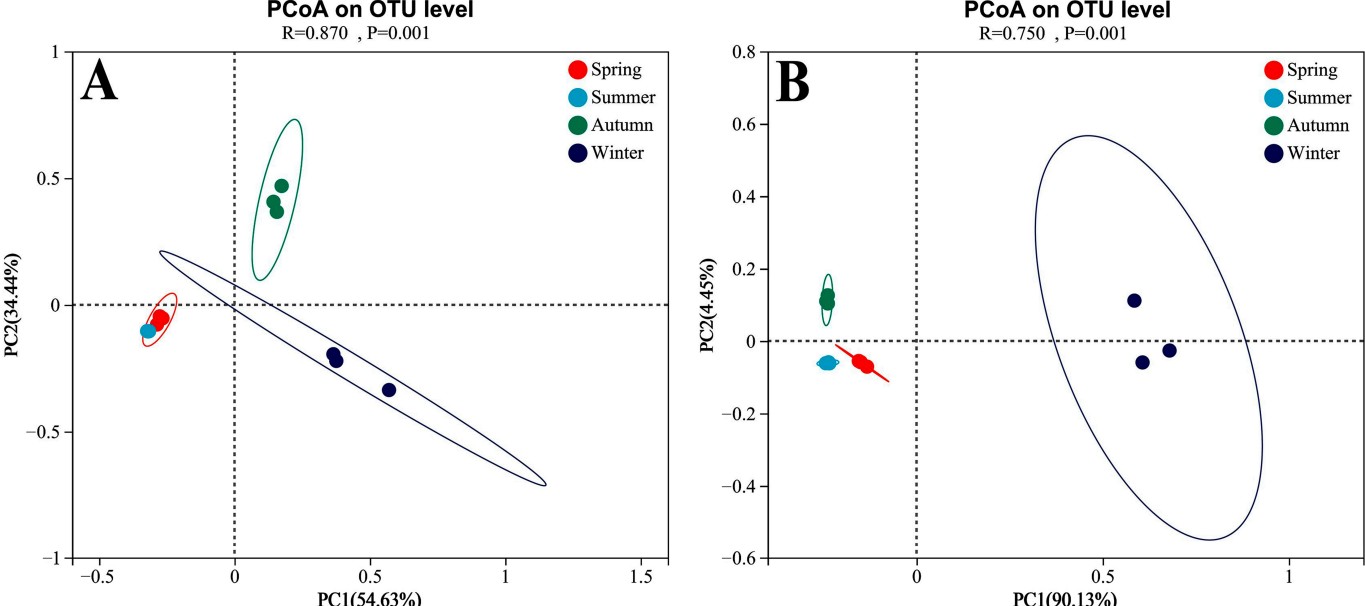

**Figure 3.** Principal-Coordinate Analysis (PCoA) of phyllosphere bacterial (**A**) and fungal (**B**) communities in nettle. Season was the sampling time. Spring, 10 May; Summer, 10 July; Autumn, 10 September; Winter, 10 November.

Figure 4A,B show the variation in phyllosphere bacterial community diversity and relative abundance in different seasons. In spring and summer, Cyanobacteria (81.29–98.12%) was the dominant phylum, whereas, in autumn and winter, Proteobacteria (24.10–98.12%) was the dominant phylum, followed by Cyanobacteria (5.75–28.08%) (Figure 4A). The distributions of the bacterial communities at the genus level are shown in Figure 4B. The differences in distributions of the bacterial genera were considerable in different seasons, and the top five most abundant genera were *norank_f_norank_o_Chloroplast*, *Sphingomonas*, *Massilia*, *unclassified_f_Oxalobacteraceae*, and *Hymenobacter*.

The relative abundances of *norank_f_norank_o_Chloroplast* were the highest in spring (81.19–88.10%), summer (96.93–98.49%), and autumn (23.12–27.10%). *Sphingomonas* was the dominant genera in winter (25.22–28.45%). In spring, excluding the dominant genera, *unclassified_f_Enterobacteriaceae* (2.78–5.99%), *unclassified_0_Enterobacterales* (1.64–3.69%), and *Enterobacter* (1.06–2.31%) also accounted for a certain proportion of the community. In summer, excluding the dominant genera, the average relative abundance of other genera was <1%. In addition, in autumn, excluding the dominant genera, there were numerous bacterial genera, such as *Enterococcus* (8.10–18.78%), *Sphingobacterium* (1.23–15.76%), *Enterobacter* (5.71–6.85%), *Ochrobactrum* (0.50–8.40%), *Bacillus* (1.46–9.64%), and *Streptomyces* (0.55–6.15%). Winter also had various bacterial genera, such as *norank_f_norank_0_Chloroplast* (5.75–28.06%), Massilia (5.23–20.28%), *unclassified_f_Oxalobacteraceae* (4.78–20.05%), *Hymenobacter* (6.57–9.89%), *Allorhizobium-Neorhizobium-Pararhizobium-Rhizobium* (1.23–2.73%), *Pseudomonas* (1.19–2.62%), and *Clavibacter* (1.14–2.24%). The average relative abundances of other genera were <2% in autumn and winter.

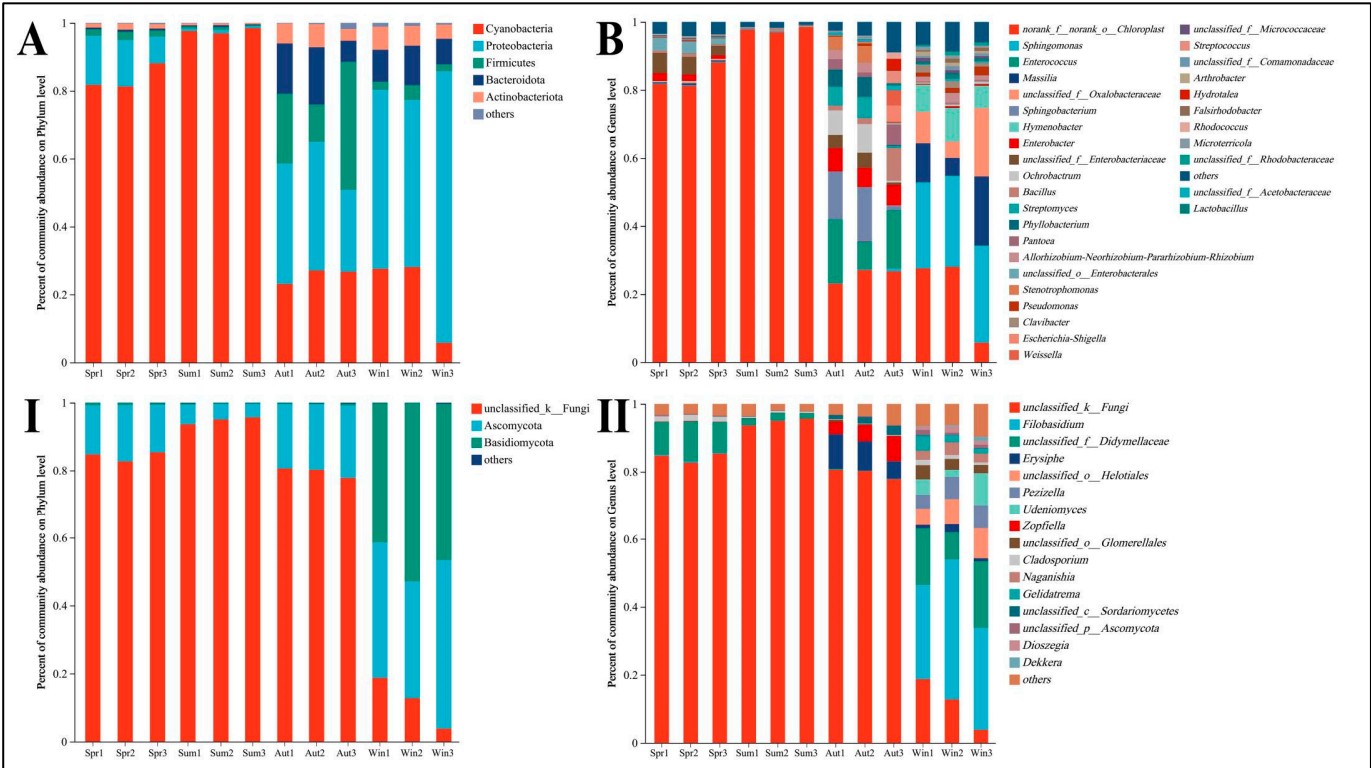

**Figure 4.** Phyllosphere microbial community composition and relative abundances based on bacterial (**A**) and fungal (**I**) phylum levels and bacterial (**B**) and fungal (**II**) genus levels in nettle. Spr, 10 May; Sum, 10 July; Aut, 10 September; Win, 10 November. The measurements were repeated three times per season.

Figure 4I,II show the phyllosphere fungal community structures and relative abundances in different seasons. The relative abundance of unclassified_k_Fungi (77.82–95.67%) was the highest from spring to autumn, followed by Ascomycota (4.00–21.34%). In winter, Basidiomycota was the dominant phylum (41.27–52.82%), followed by Ascomycota (34.32–49.55%), and unclassified_k_Fungi (3.84–18.76%) (Figure 4I). The distributions of the fungal communities at the genus level are shown in Figure 4II. The fungal genera had considerable differences in different seasons, and the top 5 most abundant genera were *unclassified_k_Fungi*, *Filobasidium*, *unclassified_f_Didymellaceae*, *Erysiphe*, and *unclassified_o_Helotiales*. In spring, the dominant fungal genera were *unclassified_k_Fungi* (77.82–95.67%), followed by *unclassified_f_Didymellaceae* (9.02–11.63%), and *Cladosporium* (1.44–1.73%); in summer, the dominant genera were *unclassified_f_Didymellaceae* (1.48–2.11%); and in autumn, the dominant genera were *Erysiphe* (5.15–10.20%) and *Zopfiella* (3.66–7.63%). The average relative abundances of other genera were < 1%. In winter, *Filobasidium* (27.67–41.14%) was the dominant genus, followed by *unclassified_f_Didymellaceae* (8.05–19.64%), *unclassified_k_Fungi* (3.84–18.76%), *unclassified_o_Helotiales* (4.90–8.88%), *Pezizella* (4.16–6.94%), *Udeniomyces* (2.06–9.46%), *unclassified_o_Glomerellales* (2.43–4.18%), *Naganishia* (2.73–3.79%), *Gelidatrema* (1.46–4.39%), *Erysiphe* (0.83–2.34%), and the average relative abundances of the other genera were <1%.

Linear Discriminant Analysis Effect Size (LefSe) analysis was performed to further explore the variations in the bacterial (Figure 5A) and fungal (Figure 5B) community structure among the different seasons. The relative abundance of *norank_f_norank_o_Chloroplast* was significantly higher in spring and summer than in autumn and winter ($p < 0.05$). The relative abundances of *Sphingobacterium*, *Enterococcus*, *Enterobacter*, and *Ochrobactrum* were the highest in autumn ($p < 0.05$), and the relative abundances of *Sphingomonas*, *Massilia*, *unclassified_f_Oxalobacteraceae*, and *Hymenobacter* were the highest in winter ($p < 0.05$). The relative abundance of *unclassified_k_Fungi* was the lowest in winter ($p < 0.05$), but the

relative abundances of *Filobasidium*, *unclassified_o_Helotiales*, *Pezizella*, *Udeniomyces*, *Naganishia*, and *Gelidatrema* were significantly higher than those in other seasons ($p < 0.05$). In addition, the relative abundance of *Erysiphe* was the highest in autumn, whereas those of *unclassified_f_Didymellaceae* and *Cladosporium* were the highest in spring and winter ($p < 0.05$).

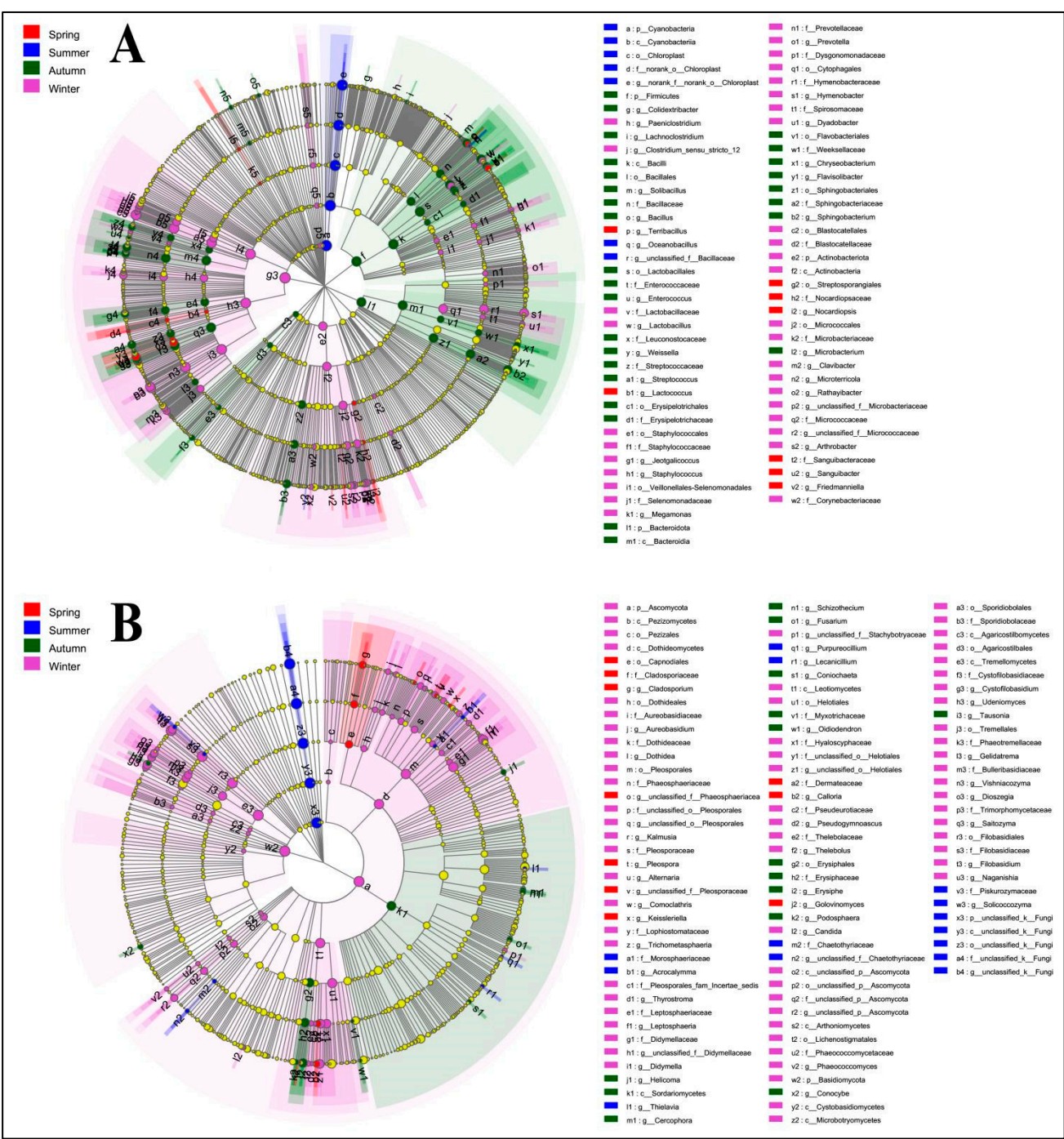

**Figure 5.** Comparison of phyllosphere bacterial (**A**) and fungal (**B**) variations using Linear Discriminant Analysis (LDA) Effects Size (LefSe) analysis in nettle. LDA score (log10) was greater than 3.0.

### 4.3. Relationship between Phyllosphere Microbial Composition and Amino Acids

Redundancy Analysis (RDA) results showed that AAs play a key role in the distribution of bacterial (Figure 6A) and fungal (Figure 6B) community distribution. The two principal components explained the relationship between bacteria (95.72%), fungi (97.08%), and AA. *Norank_f_norank_o_Chloroplast* and *unclassified_k_Fungi* had the highest correlation with AA, followed by *Sphingomonas* and *Filobasidium*. AAs that had a considerable, important influence on the distribution of phyllosphere microbial community distribution nettle were proline (bacteria, $r^2 = 0.88$, $p < 0.01$; fungi, $r^2 = 0.92$, $p < 0.01$), tyrosine (bacteria, $r^2 = 0.97$, $p < 0.01$; fungi, $r^2 = 0.86$, $p < 0.05$), serine (bacteria, $r^2 = 0.79$, $p < 0.05$; fungi, $r^2 = 0.84$, $p < 0.01$) and phenylalanine (bacteria, $r^2 = 0.74$, $p < 0.01$; fungi, $r^2 = 0.83$, $p < 0.01$).

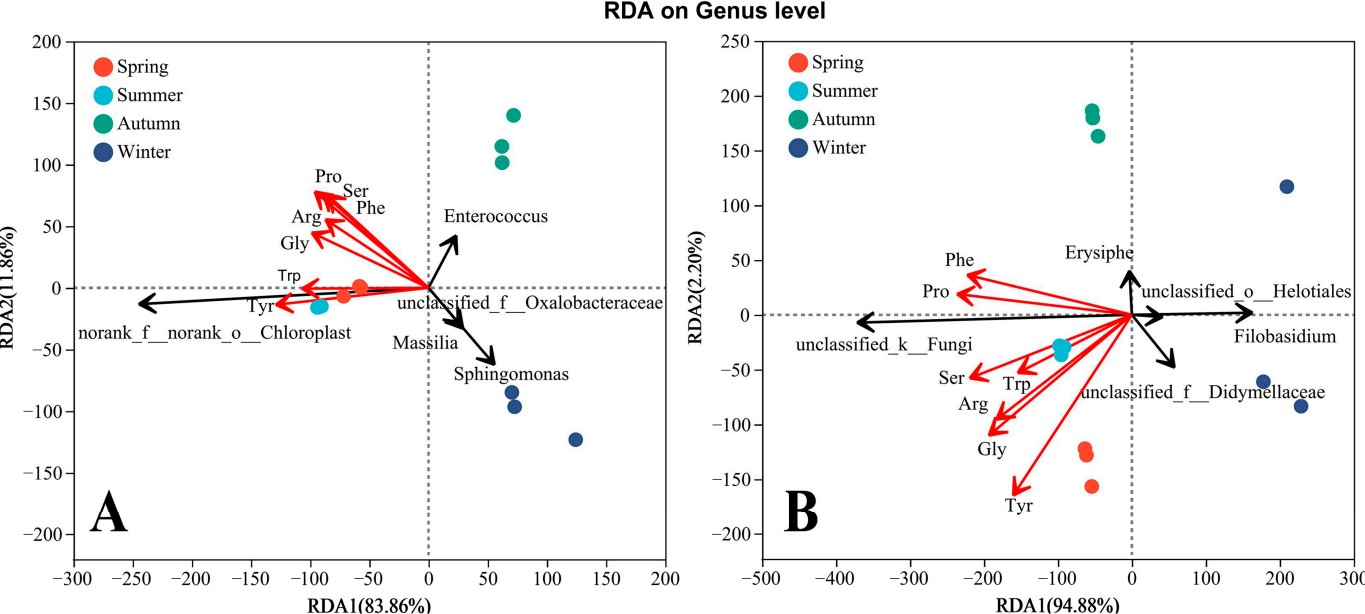

**Figure 6.** Redundancy analysis (RDA) of the relationship between the phyllosphere bacterial (**A**) and fungal (**B**) relative abundances and amino acids. The environmental factors with $p > 0.05$ or Variance Inflation Factor (VIF) > 10 were screened. Gly, glycine; Ser, serine; Arg, arginine; Pro, proline; Phe, phenylalanine; Tyr, tyrosine; Trp, tryptophan. Red arrows represent AA, and black ones represent microbial.

A Spearman's correlation heatmap was used to illustrate the correlations between AA contents and the relative abundance of bacteria (Figure 7A) and fungi (Figure 7B) at the genus level. The proline and phenylalanine contents were negatively and significantly correlated with *Sphingomonas* (r = −0.85 and −0.80), *Hymenobacter* (r = −0.85 and −0.83), *Massilia* (r = −0.86 and −0.83), *unclassified_f _Oxalobacteraceae* (r = −0.87 and −0.81), and *Pezizella* (r = −0.83 and −0.78), *unclassified_o_Glomerellales* (r = −0.67 and −0.78), *unclassified_f_Didymellaceae* (r = −0.69 and −0.64), *Udeniomyces* (r = −0.70 and −0.70), *Filobasidium* (r = −0.76 and −0.82), *unclassified_o_Helotiales* (r = −0.81 and −0.80). Tyrosine contents were negatively and significantly correlated with *Ochrobactrum* (r = −0.63), *Enterococcus* (r = −0.75), *Sphingobacterium* (r = −0.64), *Sphingomonas* (r = −0.76), and *Pezizella* (r = −0.58) relative abundance. Serine contents were negatively and significantly correlated with the relative abundance of *Ochrobactrum* (r = −0.59) and *Pezizella* (r = −0.59). However, notably, AA contents were positively correlated with *norank_f_norank_o_Chloroplast* and *unclassified_k_Fungi* relative abundance. In particular, *norank_f_norank_o_Chloroplast* and tyrosine (r = 0.90), tryptophan (r = 0.85) exhibited significant and positive correlation; and *unclassified_k_Fungi* had positive and significant correlation with other AAs, excluding serine and arginine. Only the top 10 most abundant genera were analyzed.

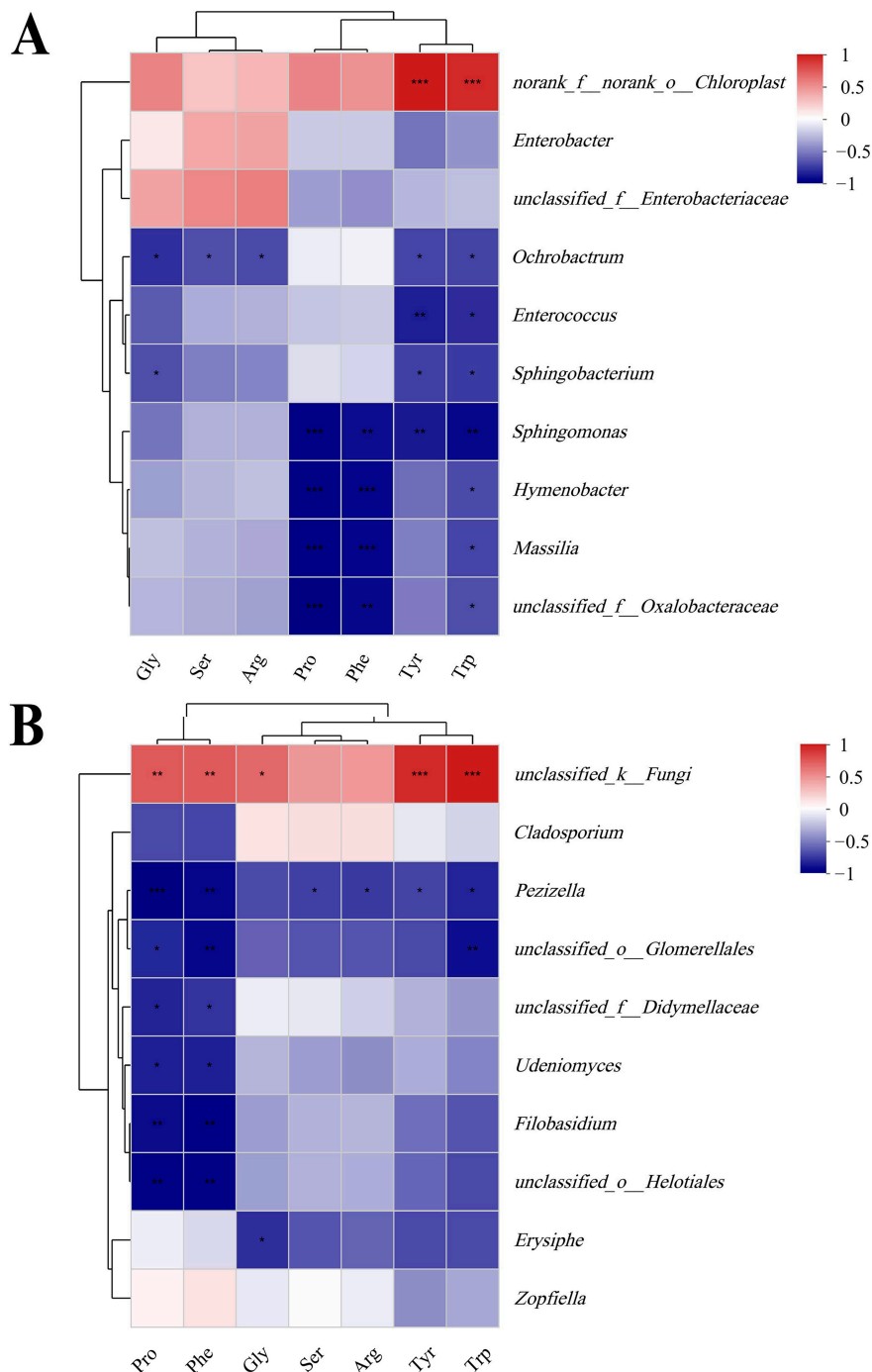

**Figure 7.** Spearman's correlation heatmap of the abundance of bacterial (**A**) and fungal (**B**) genera and amino acids in nettle leaves. * $p < 0.05$, ** $p < 0.01$ and *** $p < 0.001$. Gly, glycine; Ser, serine; Arg, arginine; Pro, proline; Phe, phenylalanine; Tyr, tyrosine; Trp, tryptophan.

### 4.4. Functional Prediction of Phyllosphere Microbes

Phylogenetic Investigation of Communities by Reconstruction of Unobserved States (PICRUSt2) analysis was used to predict phyllosphere bacteria function (Figure 8A). Following analyses of the functional abundance and composition of phyllosphere bacteria based on the Kyoto Encyclopedia of Genes and Genomes (KEGG) pathways under different seasons, the authors observed that seasons did not significantly affect the composition and functional abundance. The major enrichment pathways included metabolic pathways (15.83–18.79% relative abundance), biosynthesis of secondary metabolites (7.24–8.65% rela-

tive abundance), biosynthesis of antibiotics (5.15–5.23% relative abundance), and microbial metabolism in diverse environments (3.93–5.11% relative abundance).

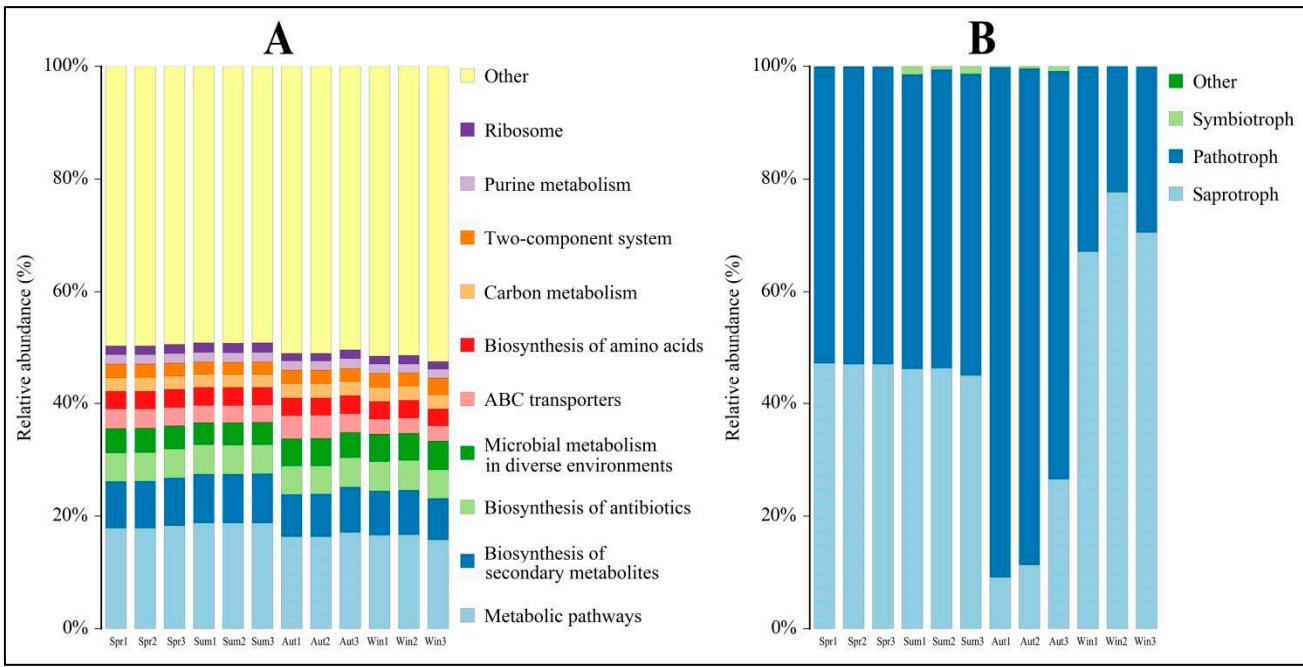

**Figure 8.** Functional prediction of phyllospheric bacterial (**A**) and fungal (**B**) communities. PICRUS2 (KEGG pathway) and FUNGuild were used for bacterial and fungal community function prediction.

FUNGuild was used to classify the functions of phyllospheric fungi (Figure 8B). According to the different nutrition types, pathotrophs, and saprotrophs in the four seasons accounted for >97% of all fungal OTUs. The relative abundance of pathotrophs (relative abundance was 52.35–53.59%) and saprotrophs (relative abundance was 45.00–47.09%) both showed high levels in spring and summer, but pathotrophs were dominant in autumn (72.50–90.64% relative abundance), whereas saprotrophs were dominant in winter (67.11–77.65% relative abundance). In addition, a few symbiotrophs were observed in summer (<2% relative abundance).

## 5. Discussion

AA contents affect forage nutritional value directly and enhance tolerance to abiotic stress during forage growth [29]. In the present study, AA contents in spring and summer were higher than those in autumn and winter, which may be attributed to abiotic environmental stress. Previous studies have shown that high temperatures and drought increase AA contents in plants [30,31]. Due to the local drought in summer, the contents of AAs such as proline, tyrosine, phenylalanine, threonine, and tryptophan were accumulated to enhance plant drought tolerance [32]. However, some AAs with high contents in spring, such as glycine and serin, are not only affected by the environment but also play a major role in plant growth [33].

Microbial activity and diversity are affected by temperature and water; thus, microbial community structures respond to seasonal changes [34]. According to PCoA in Figure 3, the local environment also affected phyllosphere microbial richness and diversity in the present study. In addition, seasonal changes make plants susceptible to harmful microbes [35]. The relative abundance of Cyanobacteria decreased in autumn and winter, and Proteus, Firmicum, Bacteroides, and Actinomycetes abundance increased, which indicated that with a decrease in nettle photosynthesis in autumn and winter, bacteria involved in photosynthesis were inhibited [36]; and various bacteria helped nettle resist drought, cold, and other adverse environmental conditions [37]. Previous studies have shown that *Sphingomonas*

and *Sphingobacterium* could promote plant growth and environmental stress tolerance [38]. In the present study, *Sphingomonas* was the dominant genus, with the highest relative abundance (26.77%) in winter. However, *Sphingobacterium* had the highest relative abundance (10.33%) only in autumn. The capacities of the two genera to facilitate plant growth and stress tolerance could be different.

Previous studies have shown that some *Sphingomonas* spp. strains can fix nitrogen [39], and some could protect plant leaves from pathogenic infections [19]. Such factors could be the main pathways via which nettle resists abiotic and biotic stress. *Massilia* is mostly found in extreme environments. It synthesizes multiple secondary metabolites and enzymes and has high cold tolerance, phosphorus dissolution function, and other functions [40,41]. *Hymenobacter* is widely distributed and is primarily found in extreme environments; it has high radiation and low-fertility tolerance characteristics [42]. Similarly, in the present study, *Massilia* and *Hymenobacter* had certain activities, and their relative abundances (12.33% and 8.14%, respectively) were the highest in winter. According to the results above, the area's large temperature difference and harsh environment could be the main reasons for the higher relative abundances of *Sphingomonas*, *Sphingobacterium*, *Massilia*, and *Hymenobacter*.

In the case of fungi, previous studies have shown that *Filobasidium* is a typical basidiomycetes genus, which easily attaches to fruit surfaces, and its abundance increases with an increase in fruit maturity [43]; furthermore, it exhibits a certain degree of radiation tolerance [44]. In addition, some of the genus *pezizella* easily form mycorrhiza with plants to promote nutrient absorption, which could facilitate plant survival in winter [45]. Similarly, in the present study, *Filobassidium* and *Pezizella* relative abundances (32.92% and 5.89%, respectively) were the highest in winter. Previous studies have shown that the secondary metabolites of *Zopfiella*, an endophytic fungus, have anti-tumor and antibacterial functions and enhance host plant stress resistance [46,47]. Since fungal pathological activity increases in autumn, *Zopfiella* relative abundance increased (5.28%) to enhance nettle stress resistance. Furthermore, the highest relative abundance of *Erysiphe* observed in autumn (8.00%) is a key factor associated with powdery mildew [48].

According to the RDA results for nettle phyllosphere microbes and AA contents, unclassified bacteria and fungi were the most affected by season change, followed by *Sphingomonas* and *Filobasidium*. In the present study, proline, tyrosine, serine, and phenylalanine influenced phyllosphere microbial community distribution in nettle. The winter certainly impacted both bacterial and fungal communities, especially for fungi in nettle. As shown in Figure 6, *Sphingomonas* and *Filobasidium* were important nettle factors during winter. These two microbes showed a negative correlation with AA; fungi showed fewer correlations with AA compared with bacteria (Pro, Phe, Tyr, and Trp for *Sphingomonas*, Pro, and Phe for *Filobasidium*). *Sphingomonas* can use various organic compounds, including hydrogen, carbon dioxide, and some sugars, in addition to AA, to support autotrophic or heterotrophic activity, and can grow and reproduce under extremely oxygen-deficient or harsh conditions [49], which explains the significant increase in *Sphingomonas* relative abundance in the present study in winter, because harsh conditions such as cold winters inhibit the growth of other microorganisms, whereas *Sphingomonas* could survive in such environments.

Previous studies have shown that *Sphingomonas* inoculation can reduce plants' hydrogen peroxide, malondialdehyde, and proline contents, increase antioxidant enzyme activity, and regulate the ascorbate–glutathione cycle [50,51]. According to the results of correlation analysis in the present study, *Sphingomonas* had negative and significant correlations with proline content, indicating that *Sphingomonas* had a similar role in reducing proline content. In addition, *Sphingomonas* had significant and negative correlations with tyrosine, serine, and phenylalanine contents, which might be related to the utilization of aromatic compounds by *Sphingomonas*, which has excellent aromatic compound degradation qualities [52], and it can exploit various compounds to grow and reproduce under adverse conditions [49]. Similarly, the benzene ring structure exists in the structures of the three AAs. *Sphingomonas* could promote seed maturation, in turn enriching *Filobasidium* on the

plant surface, and its relative abundance increased with plant maturity [43]. In addition, the northern hemisphere experiences relatively strong sun radiation in winter [53], and the increase in the relative abundance of *Filobasidium* could be associated with the enhancement of nettle radiation resistance [44].

In the present study, PICRUSt2 was used to predict phyllosphere microbial function in nettle in different seasons based on KEGG pathways. According to the results of the present study, the bacterial community in the leaf zone of nettle actively participates in basic metabolic processes, consistent with previous research findings [54]. Although species composition differed between autumn and winter, their KEGG pathways were similar, indicating that the phyllosphere bacteria have similar functions. Undefined bacteria accounted for a large proportion of the community, and their function should be explored in further studies. FUNGuild prediction results showed that the aggregated fungi are mainly pathotrophic and saprophytic fungi in spring and summer. The relative abundances of pathotrophic fungi increased in autumn, whereas the relative abundances of saprophytic fungi increased in winter. Considering the fungal species compositions, the high abundances of *Erysiphe* in autumn and Filobasidium in winter could be the main reason for the difference [48].

## 6. Conclusions

AA contents in *U. cannabina* generally decreased from spring to winter. Notably, AA contents were generally positively correlated with the abundance of unidentified microbes but negatively correlated with most identifiable microbes. In addition, *U. cannabina* is easily infected by *Erysiphe*, which often occurs in autumn and is potentially associated with decreased glycine content in autumn and winter. AA contents of *U. cannabina* varied significantly with seasons, and seasons and AA contents affected the microbial community structure in the leaves significantly. Proline, tyrosine, serine, and phenylalanine play key roles in phyllosphere microbial community structure in nettle leaves. The present results provide a basis for further studies of the relationship between foliar microbes, AA contents, and plant diversity.

**Author Contributions:** S.J. and Y.C. (Yongcheng Chen) wrote the manuscript and performed the partial experiments. R.H. revised the manuscript and supervised the project. Y.C. (Yuxin Chai) performed the sample collection and the partial experiments. F.Z. and C.M. were responsible for designing the experiments, formal analysis, project administration, and funding acquisition. All authors contributed to manuscript revision, read, and approved the submitted version. All authors have read and agreed to the published version of the manuscript.

**Funding:** This work was supported by the National Natural Science Foundation of China (NSFC) [grant number 32060399] and the China Agriculture Research System of the Ministry of Finance (MOF) and the Ministry of Agriculture and Rural Affairs (MARA) [grant number CARS].

**Institutional Review Board Statement:** Not applicable.

**Data Availability Statement:** The data that support this study are available in the article. The raw data in this study are available upon request from the corresponding author.

**Conflicts of Interest:** The authors declare that the research was conducted without any commercial or financial relationships that could be construed as a potential conflict of interest.

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
