# Peer review of "Study on Phyllosphere Microbial Community of Nettle Leaf during Different Seasons"

_agriculture, doi:10.3390/agriculture13061271_

Round 1
Reviewer 1 Report
In this work, the seasonal influence on the content of amino acids and microbial (bacterial and fungal) community structure of nettle plants was examined. In my opinion, the study was well conducted and presented in the manuscript. Minor objections must be corrected before acceptance.
Line 13: Please, define the abbreviation „AA“ in the Abstract.
Table 1: Please, show results as mean ± standard deviation and define the abbreviation SEM.
Line 251-254: Should this disclaimer be here in the figure caption?
Author Response
Comments and Suggestions for Authors
In this work, the seasonal influence on the content of amino acids and microbial (bacterial and fungal) community structure of nettle plants was examined. In my opinion, the study was well conducted and presented in the manuscript. Minor objections must be corrected before acceptance.
Q1: Line 13: Please, define the abbreviation „AA“ in the Abstract.
Respond:Added AA, meaning amino acids
Q2: Table 1: Please, show results as mean ± standard deviation and define the abbreviation SEM.
Respond:SEM meaning standard error of mean,
Q3: Line 251-254: Should this disclaimer be here in the figure caption?
Respond:It has been revised.

Reviewer 2 Report
The manuscript entitled ‘Study on phyllosphere microbial community of nettle leaf during different seasons’ is interesting manuscript that described the variation of bacteria and fungi at four stages along with variation of AA.
However following points needs to be addressed.
Thorough English revisions is needed. In addition to grammatical mistakes, at many points’ sentences are too large to understand the main point.
Abstract lacks the answer to question ‘why phyllosphere microbes are important’. Kindly restructure the abstract.
L13- write AA full form.
L21: this two microbes were both most important factors, correct the sentence. Similarly check throughout manuscript for other language errors such as L94, L103 etc
Introduction lacks previous reports of such studies if any in same plant or different plants and what this study added to previous literature.
Collection time would have impact over microbial growth and AA composition. Current collection time is peak sunshine and probably most of the microbes are less dense during summer at this time. Any reason of choosing 11:30-12 AM?
PCR conditions were same for bacteria and fungi?
Figure 3: three dots of same color are replicates? Technical or biological?
Correlation with season, temperature and humidity are two important factors. Graphs of temperature and humidity of collection stages must be added and must be discussed in context of microbial growth and AA variation.
Improve figure 4 for better understanding. Spring , summer , autumn and winter notions are used as well as months are also described. Here both are depicted in legends but
Kindly check caption of figure 5. Unnecessary text there.
Linear Discriminant Analysis Effect Size, kindly explain significance of this test. And how it complement other tests if any.
Figure 6: Red arrows represent AA? And black represent both fungi and bacteria? Mention this in legend.
Rephrase sentence stating at L329, L348,
According to KEGG functional prediction, there is no seasonal variation in terms of function though communities are different. Plant needs vary according to season, how would you justify this functional stability in terms of plant microbe interaction? In terms of AA diversity? Discuss in terms of nettle life cycle.
Improve conclusion by adding above points. Add significant and relevant future prospective of current study.
Extensive English editing needed.
Author Response
Comments and Suggestions for Authors
The manuscript entitled ‘Study on phyllosphere microbial community of nettle leaf during different seasons’ is interesting manuscript that described the variation of bacteria and fungi at four stages along with variation of AA.
However following points needs to be addressed.
Thorough English revisions is needed. In addition to grammatical mistakes, at many points’ sentences are too large to understand the main point.
Q1:Abstract lacks the answer to question ‘why phyllosphere microbes are important’. Kindly restructure the abstract.
Respond: We rewrite the first sentence as follow; Phyllosphere microbes are important as they have living conditions similar to those of the aboveground parts of host plants. Exploring amino acid (AA) and microorganisms can further understand the growth of plants in different seasons. . The present study investigated the content of AA and phyllosphere microbes structure of nettle plants in different seasons.
Q2:L13- write AA full form.
Respond:Added AA, meaning amino acids
Q3. L21: this two microbes were both most important factors, correct the sentence. Similarly check throughout manuscript for other language errors such as L94, L103 etc
Introduction lacks previous reports of such studies if any in same plant or different plants and what this study added to previous literature.
Respond:We added some sentences to the introduction, as follow
Line59: There had co-occurrence networks between rhizosphere and phyllospheric microbial of plants. Consequently, phyllospheric microbes play a key role, mainly because they have living conditions similar to those of the aboveground parts of host plants and networks with others microbial from different part of plants
Lin 68: So far, studies focus on nettle growth such as physiological and photosynthetic charac-teristics, none of them had studies on microbial community of nettle.
Q4. Collection time would have impact over microbial growth and AA composition. Current collection time is peak sunshine and probably most of the microbes are less dense during summer at this time. Any reason of choosing 11:30-12 AM?
Respond: We focus on the effect of stress of environment such as temperature on microbial activity of nettle leaf. So we choose the 11:30-12:00 am for each day to collected samples, cause in that period the stress was highest
Q5. PCR conditions were same for bacteria and fungi?
Respond: PCR reaction system are different but PCR reaction parameters are same between bacteria and fungi. The difference as follow “TransStart Fastpfu DNA Polymerase, 20 μl reaction system” for bacteria and “TaKaRa rTaq DNA Polymerase, 20 μl reaction system” for fungi, respectively:
Table 1 Primer design for bacteria and fungi
Sequencing region |
Primer name |
Sequences |
338F_806R |
338F |
ACTCCTACGGGAGGCAGCAG |
806R |
GGACTACHVGGGTWTCTAAT |
|
ITS1F_ITS2R |
ITS1F |
CTTGGTCATTTAGAGGAAGTAA |
ITS2R |
GCTGCGTTCTTCATCGATGC |
Table 2 PCR reaction system for bacteria and fungi
Bacteria |
Fungi |
||
Name |
Volum |
Name |
Volum |
5×FastPfu Buffer |
4 μl |
10× Buffer |
2 μl |
2.5 mM dNTPs |
2 μl |
2.5 mM dNTPs |
2 μl |
Forward Primer(5 μM) |
0.8μl |
Forward Primer(5 μM) |
0.8 μl |
Reverse Primer(5 μM) |
0.8μl |
Reverse Primer(5 μM) |
0.8 μl |
FastPfu Polymerase |
0.4μl |
rTaq Polymerase |
0.2μl |
BSA |
0.2μl |
BSA |
0.2 μl |
Template DNA |
10 ng |
Template DNA |
10 ng |
Figure 3: three dots of same color are replicates? Technical or biological?
Respond: three dots of same color are biological replicates.
Q6. Correlation with season, temperature and humidity are two important factors. Graphs of temperature and humidity of collection stages must be added and must be discussed in context of microbial growth and AA variation.
Respond:We focused on the seasonal effects on microbial structure of nettle leaf, in this condition, many factors are variable such as humidity and temperature. We just want introduce the microbial structure changed among seasonal, give the whole picture for microbial structure during different seasonal, the specific factors are not discussion. We added the results and methods for temperature during period of sample collected. As follow;
Respond:Lin 109 , Temperature and humidity during sampling times were obtained by use temperature and humidity recording instrument (testo 625, china) for interval of 5 minutes during 11: 30-12: 00AM, which were 20.3℃ and 3.1% for May 10th (spring), 28.5℃ and 26.93% for July 10th (summer), 16.32℃ and 10.19% for September 10th (autumn), and 1.27℃ and 62.14% for November 10th.
Q7. Improve figure 4 for better understanding. Spring , summer , autumn and winter notions are used as well as months are also described. Here both are depicted in legends but
Q8. Kindly check caption of figure 5. Unnecessary text there.
Respond: checked
Q9. Linear Discriminant Analysis Effect Size, kindly explain significance of this test. And how it complement other tests if any.
Respond: Linear Discriminant Analysis Effect Size could help us better understand the significant difference of microbial between different seasons.
Q10. Figure 6: Red arrows represent AA? And black represent both fungi and bacteria? Mention this in legend.
Respond: checked, added Line 290, Red arrows represent AA and black represent micaobial.
Q11.Rephrase sentence stating at L329, L348
Respond: Changes as follow;
Line 329,The relative abundance of pathotrophs (relative abundance was 52.35%–53.59%) and saprotrophs (relative abundance was 45.00%–47.09%) were both showed high level in spring and summer, but pathotrophs were dominant in autumn (72.50%–90.64% relative abundance), whereas saprotrophs were dominant in winter (67.11%–77.65% relative abundance). In addition, a few symbiotrophs were observed in summer (< 2% relative abundance).
Line 348, According to PCoA in Fig.3, in the present study, phyllosphere microbial richness and diversity were affected by seasonal changed.
Q12.According to KEGG functional prediction, there is no seasonal variation in terms of function though communities are different. Plant needs vary according to season, how would you justify this functional stability in terms of plant microbe interaction? In terms of AA diversity? Discuss in terms of nettle life cycle.
Respond:As mention, we just want discussion of functional prediction of Phyllosphere Microbes. On the noe hand, the KEGG data base is limit, not for all microbial. The functional prediction based on the sequences of microbial, in that condition, the functional probably be the same even compostion of microbial is difference.
Improve conclusion by adding above points. Add significant and relevant future prospective of current study.

Reviewer 3 Report
This work is interesting and should be accepted for publication in Agriculture after proofreading. However, the entire manuscript should be re-read and many typographical errors should be corrected. The methodology requires considerable detail. We learn about many methodological elements only in "Results". The discussion needs to be supplemented with numerous elements, some of which are indicated in Remarks. In the discussion, there is too little critical analysis of one's own results and attempts to explain the relationships found. There is no mention anywhere in the text that the bacterial and fungal community found concerns the surface of leaves and endophytic occurrence inside the leaf tissue. The role of these microorganisms varies depending on where they occur, on the surface or as endophytes.
Remarks
In many places throughout the manuscript, spacing between different words or characters is not maintained. This should be corrected.
Line 13 it should be: Amino acids (AA) insted of AA
Line 25 Pathogenic fungi and others, can we talk about pathogenic fungi, if there is no mention of disease symptoms at all in the text
Line 31 Urtica – throughout the manuscript proper names of plants, fungi and bacteria should be written in italic
Line 44 it should be variation [5]. Similarly, line 51 - plants[9,10]., line 52 - imbalance[11], and in several other places throughout the manuscript
Line 70 - please, define how phyllosphere is understood in this work. In the literature, the phyllosphere is defined in various ways. This is often understood as the surface of leaves, considered as a habitat for microorganisms. The applied methodology makes it impossible to define the phyllosphere in this way - according to the applied methodology, the occurrence of epiphytes and edophytes in leaves was assessed ??.
Line 78 m a.s.l. ??
Line 85 it should be subsp. sulcata, instead of ‘subsp. Sulcata,’
Line 93 must be stated whether nettle leaves were symptomless in all seasons? This aspect should also be mentioned in the discussion. This is important in the context of stating, for example, Erysiphe (mildews causal fungi)
Line 95 should be explained why the leaf surfaces were scraped (we can guess but it should be written)
Line 112 specify which nettle samples were analyzed in this way (see Line 95 – 99)
Line 149 P < 0.05 – in other places it is p < 0.05 . The notation should be unified
Line 164 ‘OTUs based on 97% sequence identity’ - such data should be provided in Material and Methods. Literature should be cited. It is necessary to discuss in Discussion how such methodical procedure could affect the obtained results
Line 167 define in Materials and Methods the term used for fungi and bacteria - diversity and richness. The term Alpha diversity is used for the first time in Figure 2; relative abundance - used in Figure 4. This should all be listed and defined in Material and Methods
Line 192 '5 most abundant species' - names listed represent genera, not species
Line 192 norank_f_nor-192 ank_o_Chloroplast - this group should be described in more detail
Line 261 consider revising this sentence
Line 207 numerous bacterial species, such as …. Genera are listed, not species
Line 292 you get results: unclassified_k_Fungi, unclassified_o_Glomerellales, unclassified_o_Helotiales, unclassified_f_Didymellaceae, also other taxa are identified only to genus level - so how is it possible to determine if it is a pathogen or a saprotroph without knowing the species of the fungus. Of course, the program can calculate general data, but what is the logic and sense of it? The same goes for bacteria.
Line 339 spp. - in this case it should be rather species
Line 354 ‘Pezizella easily forms mycorrhiza’ - all Pezizella species or a specific species of Pezizella? - in many places the text is imprecise
Line 359-360 this text is unclear; 'Since fungal pathological activity increases in autumn, Zopfiella....' what is this statement based on?
Line 361-362 this text is unclear - Erysiphe is the causal agent of powdery mildew
Line 368 improtant – rather important
Line 368 this text is unclear,
Line 390 – 391 consider revising this sentence
Line 394 – 395 'According to the results of the present study, the bacterial community in the leaf zone of nettle actively participate in basic metabolic processes' - please provide specific data what do the authors mean, whether such studies were presently conducted?
Line 398 "indicating that the phyllosphere bacteria have similar functions" - this text is unclear
Line 401 "are mainly pathotrophic and saprophytic vegetative fungi" - this is false content - it does not include endophytic fungi; besides - what are vegetative fungi ?
- Generally, in the Discussion, many expressions are vague, imprecise, superficial.
References : also in this chapter proper names of plants and fungi should be written in italic
Author Response
Comments and Suggestions for Authors
This work is interesting and should be accepted for publication in Agriculture after proofreading.However, the entire manuscript should be re-read and many typographical errors should be corrected. The methodology requires considerable detail. We learn about many methodological elements only in "Results". The discussion needs to be supplemented with numerous elements, some of which are indicated in Remarks. In the discussion, there is too little critical analysis of one's own results and attempts to explain the relationships found. There is no mention anywhere in the text that the bacterial and fungal community found concerns the surface of leaves and endophytic occurrence inside the leaf tissue. The role of these microorganisms varies depending on where they occur, on the surface or as endophytes.
Q1: In many places throughout the manuscript, spacing between different words or characters is not maintained. This should be corrected.
Respond:We revised such questions and checked similar places in the manuscript.
Q2: Line 13 it should be: Amino acids (AA) insted of AA
Respond:The modification has been completed.
Q3: Line 25 Pathogenic fungi and others, can we talk about pathogenic fungi, if there is no mention of disease symptoms at all in the text
Respond:Pathogenic fungi are known from the function prediction of fungi. According to the existing research, they may cause diseases according to certain credibility and are classified as pathogenic fungi.
Q4: Line 31 Urtica – throughout the manuscript proper names of plants, fungi and bacteria should be written in italic
Respond:We have revised the italicization of the corresponding proper nouns.
Q5: Line 44 it should be variation [5]. Similarly, line 51 - plants[9,10]., line 52 - imbalance[11], and in several other places throughout the manuscript
Respond:We made some adjustments and changed it to: In addition, seasonal changes have a great relationship with plant diseases.
Q6: Line 70 - please, define how phyllosphere is understood in this work. In the literature, the phyllosphere is defined in various ways. This is often understood as the surface of leaves, considered as a habitat for microorganisms. The applied methodology makes it impossible to define the phyllosphere in this way - according to the applied methodology, the occurrence of epiphytes and edophytes in leaves was assessed ?
Respond:The habitat composed of the above-ground effective parts of plants such as leaves, flowers, fruits, etc. is collectively called interlobar, and the microorganisms living on its surface and inside are called interlobar microorganisms. It doesn't just refer to microorganisms growing on the surface of leaves,We have indicated in the manuscript.
Q7: Line 85 it should be subsp. sulcata, instead of ‘subsp. Sulcata,’
Respond:This question has been revised in the manuscript.
Q8: Line 93 must be stated whether nettle leaves were symptomless in all seasons? This aspect should also be mentioned in the discussion. This is important in the context of stating, for example, Erysiphe (mildews causal fungi)
Respond:We added corresponding explanations to the manuscript.
Q9: Line 95 should be explained why the leaf surfaces were scraped (we can guess but it should be written)
Respond:The detection method of surface microorganisms must consider the accuracy and representativeness of sampling, and the surface of leaves is scraped off for this purpose.We revised it in the manuscript to read:For remove that influence of other factors on sample.
Q10: Line 112 specify which nettle samples were analyzed in this way (see Line 95 – 99)
Respond:The analysis sample is a sample that has been collected and stored in liquid nitrogen.
Q11: Line 149 P < 0.05 – in other places it is p < 0.05 . The notation should be unified
Respond:We have corrected this writing mistake.
Q12: Line 164 ‘OTUs based on 97% sequence identity’ - such data should be provided in Material and Methods. Literature should be cited. It is necessary to discuss in Discussion how such methodical procedure could affect the obtained results
Respond:In the 16S full-length comparison, ninety-seven percent similarity can be identified as the same species. We have changed this problem in materials and methods and added references.
Q13: Line 167 define in Materials and Methods the term used for fungi and bacteria - diversity and richness. The term Alpha diversity is used for the first time in Figure 2; relative abundance - used in Figure 4. This should all be listed and defined in Material and Methods
Respond:We corrected the related problems.
Q14: Line 192 '5 most abundant species' - names listed represent genera, not species
Respond:We corrected the relevant questions and marked them in the manuscript.
Q15: Line 192 norank_f_nor-192 ank_o_Chloroplast - this group should be described in more detail
Respond:It can't be classified at the families level, and we don't know more about it.
Q16: Line 207 numerous bacterial species, such as …. Genera are listed, not species
Respond:We corrected the related problems.
Q17: Line 292 you get results: unclassified_k_Fungi, unclassified_o_Glomerellales, unclassified_o_Helotiales, unclassified_f_Didymellaceae, also other taxa are identified only to genus level - so how is it possible to determine if it is a pathogen or a saprotroph without knowing the species of the fungus. Of course, the program can calculate general data, but what is the logic and sense of it? The same goes for bacteria.
Respond:The functions of bacteria and fungi written here are all obtained through comparison of different databases. It is a comparison of similar fungi or bacteria by analogy with some data integrated from previous studies, and the result is a prediction result, that is, what the possible functions of this fungus or bacteria are.
Q18: Line 339 spp. - in this case it should be rather species
Respond:That statement was corrected.
Q19: Line 354 ‘Pezizella easily forms mycorrhiza’ - all Pezizella species or a specific species of Pezizella? - in many places the text is imprecise
Respond:We corrected the statement here and revised it in the manuscript as follows: Some of the genus pezizella easily forms mycorrhiza with plants to promote nutrient absorption.
Q20: Line 359-360 this text is unclear; 'Since fungal pathological activity increases in autumn, Zopfiella....' what is this statement based on?
Respond:According to the fact that the relative abundance of pathological fungi increases in autumn, and Zopfiella has certain resistance, we speculate that Zopfiella relative abundance increased (5.28%) to enhance nettle stress resistance.
Q21: Line 361-362 this text is unclear - Erysiphe is the causal agent of powdery mildew
Respond:We checked and changed this article.
Q22: Line 368 improtant – rather important
Respond:We have revised the statement here in the manuscript.
Q23: Line 390 – 391 consider revising this sentence
Respond:We have revised the relevant expressions.
Q24: Line 394 – 395 'According to the results of the present study, the bacterial community in the leaf zone of nettle actively participate in basic metabolic processes' - please provide specific data what do the authors mean, whether such studies were presently conducted?
Respond:This result comes from the function prediction of bacteria, and Figure 8B shows that the function of bacteria is predicted as the basic metabolism occupies a high relative abundance, so this conclusion is drawn.We didn't do this kind of research directly. This result is based on the function prediction of bacteria in the database(PICRUSt2), which provides us with a possible function of these bacteria.
Q25: Line 398 "indicating that the phyllosphere bacteria have similar functions" - this text is unclear
Respond:According to the conclusion of bacterial function prediction (PICRUSt2), it is concluded that the KEGG pathway of phyllosphere bacteria is similar
Q26: Line 401 "are mainly pathotrophic and saprophytic vegetative fungi" - this is false content - it does not include endophytic fungi; besides - what are vegetative fungi ?
Respond:We have changed this part of the statement. Vegetative fungi refer to saprophytic vegetative types, and here is our mistake.
Q27: Generally, in the Discussion, many expressions are vague, imprecise, superficial.
References : also in this chapter proper names of plants and fungi should be written in italic
Respond:We have changed the proper names of references included in the manuscript into italics.
